# A Review on Graphene-Based Nano-Electromechanical Resonators: Fabrication, Performance, and Applications

**DOI:** 10.3390/mi13020215

**Published:** 2022-01-29

**Authors:** Yang Xiao, Fang Luo, Yuchen Zhang, Feng Hu, Mengjian Zhu, Shiqiao Qin

**Affiliations:** 1College of Advanced Interdisciplinary Studies, National University of Defense Technology, Changsha 410073, China; xiaoyang624@nudt.edu.cn (Y.X.); luofang2013@163.com (F.L.); z_yc98@nudt.edu.cn (Y.Z.); hufeng@nudt.edu.cn (F.H.); 2Hunan Provincial Key Laboratory of Novel Nano-Optoelectronic Information Materials and Devices, National University of Defense Technology, Changsha 410073, China

**Keywords:** graphene, nano-resonator, resonance frequency, sensing

## Abstract

The emergence of graphene and other two-dimensional materials overcomes the limitation in the characteristic size of silicon-based micro-resonators and paved the way in the realization of nano-mechanical resonators. In this paper, we review the progress to date of the research on the fabrication methods, resonant performance, and device applications of graphene-based nano-mechanical resonators, from theoretical simulation to experimental results, and summarize both the excitation and detection schemes of graphene resonators. In recent years, the applications of graphene resonators such as mass sensors, pressure sensors, and accelerometers gradually moved from theory to experiment, which are specially introduced in this review. To date, the resonance performance of graphene-based nano-mechanical resonators is widely studied by theoretical approaches, while the corresponding experiments are still in the preliminary stage. However, with the continuous progress of the device fabrication and detection technique, and with the improvement of the theoretical model, suspended graphene membranes will widen the potential for ultralow-loss and high-sensitivity mechanical resonators in the near future.

## 1. Introduction

A resonator measures the target parameters by detecting the frequency signal. It has the advantages of high resolution, good stability, and strong anti-interference ability [1], which is widely used in various fields. In recent years, efforts have been made to design new generation resonators with high sensitivity, fast response, and high consistency. Importantly, the miniaturization of resonators has a great significance to meet these requirements [2]. The silicon-based microstructure, which has witnessed a mature development in these years, is prone to degrading at the nanometer scale, limiting the further development of resonators in miniaturization and integration [1].

The emergence of graphene and other two-dimensional materials brings new opportunities for the miniaturization of resonators. Graphene has a high elasticity modulus, low surface density, and high surface area-to-volume ratio, leading to a significant enhancement in resonant frequency and sensitivity to external parameters, such as force, pressure, and mass [3]. However, the miniaturization also reduces the performance of the graphene devices, which is reflected in the rather low quality factor and low signal-to-noise ratio at room temperature [4]. There are still many arguments about the factor that affects the performance of graphene resonators, especially when there is a large gap between theoretical predictions and the corresponding experimental results. The research of graphene resonators is important for both fundamental research and realistic applications.

Here, we review the state-of-the-art theory and experiments of graphene resonators. Firstly, we summarize and analyze the fabrication processes of graphene and devices. Then, the electrical, optical, and piezoelectric excitation and detection schemes of the resonator are described. From theoretical analysis and experimental verification, the resonant performance including frequency tuning, loss, quality factor, and the non-linear effect are briefly introduced. Finally, the device applications of graphene resonators in the sensor, pneumatic measurement, and sensitive detection of graphene intrinsic properties are discussed.

## 2. Device Fabrication

### 2.1. Preparation of Graphene

Since graphene was discovered by mechanical exfoliation for the first time in 2004, a variety of preparation methods of graphene have been developed [5]. Among them, the graphene obtained by mechanical exfoliation has excellent electrical, mechanical, thermal, and optical properties, approaching the theoretical limits. However, the exfoliated graphene has a small size, low yield, and bad controllability on the number of layers, and it is difficult to be applied on a large scale. Chemical vapor deposition (CVD) growth is the most common method to synthesis graphene films with a large scale and controllable number of layers. Due to the catalytic property of copper and growth kinetics of graphene, most of the graphene grown on copper substrates is monolayer [6], which exhibits a large single crystal size of up to a cm scale and high mobility of up to 10,000 cm^2^·V^−1^·s^−1^ at room temperature [7]. In addition, large-scale graphene can also be synthesized by epitaxial growth on SiC substrates with a high cost and complex transfer process [8].

In resonator devices, graphene is usually placed in two ways. The first is to etch and evaporate metal electrodes followed by covering graphene on the pre-fabricated substrate through exfoliation, transfer, or direct growth. The first graphene resonator was successfully realized by exfoliating graphene onto the substrate with preset channels and electrodes, and graphene was attached to the silicon oxide wafer by van der Waals force [9]. This method is rather simple in process but with a low yield. In addition to direct exfoliation, graphene exfoliated on other substrates [10] or CVD growth graphene can also be transferred to the target substrates [11,12]. Barton et al. transferred CVD graphene grown on copper foil to suspended nitride films with holes [13]. The nitride film was suspended by back etching on the silicon substrate covered with silicon nitride and then the holes were fabricated by lithography, as shown in Figure 1a.

The second way is to place graphene on the target substrate by exfoliation, transfer, or growth firstly; then deposit metal electrodes to contact and fix the graphene; and finally suspend graphene by etching or other nano-fabrication techniques, as shown in Figure 1b. Chen et al. mechanically exfoliated graphene onto a silicon wafer covered with oxide and located the appropriate monolayer graphene through a microscope [15]. After that, they deposited the electrode and etched the silicon oxide to suspend graphene. Singh et al. deposited molybdenum on silicon oxide and patterned it, followed by growing graphene on molybdenum [16]. Then, gold was deposited by electron-beam evaporation and molybdenum was etched to form suspended graphene. In addition to directly exfoliating and growing graphene on the target substrate, the most common method is to transfer mechanically exfoliated graphene or CVD-grown graphene to the target substrate by dry transfer [14] or wet transfer approaches [17,18,19]. Compared with direct exfoliation, it can quickly and effectively locate large-size graphene with target layers on the substrate [6,20,21]. However, the transfer process is usually complicated and may degrade both the surface smoothness and mechanical properties of graphene. The fabrication processes with the above two different sequences are compared. In the first process, the graphene is finally placed on the target substrate, thus the graphene will not be exposed to polymer resistance without etching or evaporating the electrodes and has both a higher quality and integrity [22]. However, in this case, the graphene is not held by metal electrodes or other structures and thus remains unstable. One alternative is to use PMMA as a transfer agent to locate graphene to the target substrate and then overexpose the PMMA to form graphene clamping [23]. For the second process, since the graphene film is very fragile, not only the residual solvents and other polymers will contaminate the surface of graphene but also the graphene can more easily collapse or fracture in the process due to the changing surface tension. After placing graphene onto the target substrate, graphene can be etched into a specified shape by oxygen plasma etching [15,18] or focused ion-beam etching [24]. In order to avoid defects caused by etching, one can choose the target graphene with a regular shape that naturally formed by mechanical exfoliation [25].

### 2.2. Fabrication of Electrodes

The electrode of the graphene resonator can be divided into the gate electrode and source-drain electrode. The source and drain electrodes not only play the role of electrical contacts but also fix the suspended graphene film to retain the tensioned architecture. Au or other metals are deposited by plasma vapor deposition, thermal evaporation, or other means after ultraviolet lithography or electron-beam lithography. In addition, in order to improve the adhesion between the contact metal and graphene, Ti [11] or Cr [19] is usually deposited in the middle of the graphene and Au as an adhesive layer.

When the resonator is excited by electrical input, the gate electrode is required to generate electrostatic force. A simple way is making by use of the substrate directly. In this case, the heavily doped Si substrate covered with silicon oxide on top is usually working as the gate [9,15,26]. However, this method may lead to radio frequency (RF) crosstalk in the process of electrical measurement [17]. In order to reduce RF crosstalk, insulating materials such as high-resistance silicon or quartz can be used as substrates with local gates fabricated on top of them. It is worth noting that when electron-beam lithography is used for insulating materials, the charging effect on the surface will cause repulsion of the subsequent electron beam, which distorts the pattern of exposure and etching. Alternatively, the charging effect can be eliminated by using conductive resists or evaporating a metal layer on the substrate in advance [14]. Xu et al. grew 100 nm thick silicon dioxide on high-resistivity silicon wafers and then deposited local-gate electrodes on them [22]. Then, oxides were deposited for the second time to bury the local gate. After the source and drain electrodes were fabricated, the gate was finally exposed by etching silicon oxide. After the proper etching process, Zhang et al. realized the simultaneous deposition of source-drain and local-gate electrodes, simplifying the fabrication process of graphene resonators, as shown in Figure 1c [10].

### 2.3. Suspend Process

The etching process, which can be divided into dry etching and wet etching, is the most common approach used to suspend graphene. Wet etching is a pure chemical process, in which buffered oxide etchant (for example, HF/NH4F solution) is often used to etch silicon dioxide [15,17,19]. However, this method will degrade the surface morphology of silica and reduce the adhesion between the silica and resistance [27]. This problem can be solved by using an adhesion promoter such as HMDS to enhance adhesion. Another problem caused by wet etching is that the suspended graphene can easy collapse due to the changing surface tension effect. In order to protect the graphene from collapsing during wet etching, Du et al. spin-coated a PMMA layer on the transferred graphene and opened two windows on both sides of the graphene channel on the PMMA layer by photolithography [28]. In addition, following wet etching or other wet processes, graphene usually dries at a critical point [15,19,29] or infiltrates with a solution with low surface tension such as isopropanol to reduce the surface tension of graphene [26].

Compared with wet etching, dry etching has advantages of high precision, anisotropy, and fast etching speed. Reactive ion etching (RIE) combining ion bombardment and chemical reaction is the most widely used method in the dry etching of silicon dioxide [9,11,22]. Fan et al. fabricated graphene membranes with suspended silicon proof masses by dry etching of silicon and silicon oxide layers, as shown in Figure 2 [30]. The fabrication process is as follows: firstly, reactive ion etching and deep reactive ion etching are used to etch silicon dioxide and silicon on the SOI (silicon on insulator) wafer with a double-sided silicon dioxide layer to form the basic structure of the device. Then, the CVD graphene is transferred to the above substrate. Finally, combined with reactive ion etching and HF vapor etching of silicon dioxide, the silicon proof mass is released and suspended on graphene. In addition to silicon dioxide, the dry etching method can also applied to other sacrificial metal layers [16,23,31] or dielectric layers [10]. In addition, some non-etching processes can also realize the suspension of graphene. For example, the Nippon Kayaku.

SU-8 3005 (SU-8) is spun on the silicon oxide wafer and patterned by the electron beam (EB), and then the graphene is exfoliated to SU-8 followed by the SU-8 developed to suspend graphene [15,24]. Jung et al. also used development to suspend graphene by using LOR resistance [32]. Mizuno et al. used PMMA and MMA as sacrifice layers to fabricate the suspended graphene [26]. They first patterned the PMMA and MMA layers, and then sputtered NbN into contact with graphene. Finally, the sacrificial layer was removed by soaking in acetone to suspend graphene. Guan et al. applied this method to fabricate graphene resonators on flexible substrates [14]. In addition to sacrificing part of the structure, another feasible method is to preset the patterned metal layer and then graphene is transferred directly to the metal layer to obtain suspended structures [33].

## 3. Excitation and Detection

### 3.1. Excitation Schemes for Graphene Resonators

The excitation methods for graphene resonators can be summarized into electrostatic, optical, and piezoelectric excitation schemes. In the devices driven by electrostatic force, a voltage potential difference between the gate electrode and graphene is applied to generate the electrostatic force and to drive the resonance of the graphene [15,29], as shown in Figure 3a.

The gate voltage consists of two parts: a constant DC voltage maintains the tension of the graphene membrane and another RF AC voltage drives the graphene to a resonance frequency. Alternatively, when the gate electrode is grounded, the same effect can be achieved by applying the drive voltage to either the source or the drain electrode [33]. Jung et al. also found that the mixed current and the resonant signals can be effectively enhanced by fabricating two split bottom gates to form a bipolar region, as shown in Figure 3a [32].

In an optically excited device, an intensity-modulated laser spot is focused on the surface of the graphene, regulating graphene temperature via light absorption of the graphene and causing periodic contraction/expansion, as well as thus vibrations of the graphene, as shown in Figure 3b [16,34]. Optical excitation is widely used in laboratory for its convenience and efficiency because it does not require integration with electric contacts [35]. In addition, as shown in Figure 3c, Dash et al. proposed actuation and frequency tunning of graphene resonators using on-chip optical gradient force through a waveguide, which can be more efficient than conventional electrostatic and optical techniques [36]. This scheme is robust against fluctuations in the evanescent optical power and thus ensures minimal cross-talk from the optical mode to the mechanical mode.

In addition to direct excitation, parametric excitation is also widely used in electrical or optical excitation schemes [37,38], which plays a positive role in improving the quality factor, signal-to-noise ratio, and sensitivity of graphene resonators. Dolleman et al. periodically modulated the tension and stiffness of graphene monolayers through the photothermal effect, thus realizing the parametrically excited multiple resonance modes, parametric resonance, and parametric amplification of graphene resonators [39]. In the all-electric scheme, Mathew et al. realized the dynamic modulation of the multi-mode coupling of the graphene resonator through parametric excitation, which provided the feasibility for the quantum mechanical experiments at low temperature [40].

Although pristine graphene is not a piezoelectric material, in recent years, piezoelectric excitation has been successfully applied to graphene resonators by indirect driving approaches [19,41]. The process is to place the device on the piezoelectric vibrating disk and apply a voltage of a specific frequency to drive the piezoelectric disk, thus the graphene will vibrate following the piezoelectric disk, as shown in Figure 3d.

### 3.2. Detection Schemes for Graphene Resonators

The small size of the graphene resonator makes the resonance signal weak and difficult to detect, thus the detection method needs to be carefully designed and improved. Electrical and optical detection methods are the most widely used schemes to detect the resonance signals.

It is worth noting that the widely used frequency mixing technique will lead to the decrease of bandwidth and measurement speed of the graphene resonator. Alternatively, the direct RF readout technique has significant advantages in the high-speed detection for real-time applications. A common RF readout scheme for graphene resonators is to detect the RF current, combining the operation principles of graphene field-effect transistors and mechanical resonators [44]. However, reducing the parasitic capacitance, such as by adopting a local gate, to minimize the background current is required in this method [22,43]. Another way to increase the measurement speed is to couple the graphene resonator as a variable capacitor to an external circuit. For example, Song et al. coupled the graphene resonator to an LC circuit and detected the mechanical motion of graphene by the RF cavity readout technique, as shown in Figure 4b [45]. Weber et al. coupled a graphene mechanical resonator to a superconducting microwave cavity, as shown in Figure 4c. In this scenario, the principle of the vibration readout is similar to the Stokes and anti-Stokes Raman scattering [23].

The optical detection scheme, with advantages of high sensitivity and simplicity without electrical contacts, is also an important route for the effective detection of the resonance signals [46]. Interferometry is the most common method of optical detection of graphene resonators, which utilizes a laser to illuminate the suspended graphene and the substrate to form an interference cavity, as shown in Figure 3b [16,34,47]. The interference intensity is modulated by the mechanical motion of graphene and the resonant signal can be obtained by detecting the interference intensity of the laser with a photodiode. Moreover, laser doppler vibrometer (LDV) is also widely used to measure tiny displacements (0.1 pm) and velocities (25 m/s). Moreno et al. combined LDV with a vibrometer decoder for the effective signal readout of graphene resonators [41]. In addition, the vibration of graphene can also be detected by spatial imaging. Garcia-Sanchez et al. imaged the vibration eigenmodes of graphene by a scanning force microscope (SFM), as shown in Figure 4d [48]. Similarly, atomic force microscopy (AFM) was also used to spatially image the graphene resonance [46]. However, since the SFM and AFM probes cannot track high frequency vibrations, it is necessary to modulate the amplitude of the driving voltage to make graphene vibration in an envelope form.

## 4. Resonance Performance

### 4.1. Frequency Tuning

Compared to conventional Micro-Electro-Mechanical System (MEMS) resonators, graphene can withstand very high strain, suggesting an extremely wide tunable range of the resonance frequency of graphene resonators. Electrostatic tuning of the frequency can be easily achieved by applying a dynamic gate voltage. Kang et al. investigated the effect of the initial strain-induced tension and deflection-induced tension generated by a gate voltage on the resonant frequency and tuning range by molecular dynamics simulation [49,50,51]. The simulation results demonstrate that the resonance frequency of graphene increases monotonically with the increase of the electrostatic force (namely the gate voltage) and the tuning range broadens when axial strain becomes smaller. Yet, the experimental results show that increasing the electrostatic force sometimes increases the resonance frequency (spring hardening) but sometimes also decreases the frequency (spring softening), as shown in Figure 5a,b. In graphene resonators, the spring softening effect is mainly caused by non-linear electrostatic interactions, while the spring hardening effect is due to the elastic stretching of the graphene membrane [17]. The competition between the two effects may result in a ‘W’ tuning curve, as shown in Figure 5c [52]. So far, many theoretical models have been investigated to describe and understand the experimentally observed complex tuning behaviors of graphene resonators, but an accurate and universal electrostatic tuning model still needs to be developed [23,33,53,54,55].

Due to the excellent conductivity and thermal stability of graphene, Ye et al. achieved frequency tuning by Joule heating of graphene [56]. The current flow-induced Joule heating effect causes thermal stress due to the negative thermal expansion coefficient of graphene. Therefore, the additional tension in the suspended part of graphene results in the increase of the resonance frequency, as shown in Figure 5d. The tuning formula can be expressed as follows:(1)$$f_{0}=\frac{2.404}{\pi d}{\left(\frac{\gamma_{300K}-E_{Y}t\int_{300}^{T_{avg}}\alpha \left(T\right)dT}{\rho t}\right)}^{1/2}$$
where $\gamma_{300K}$ represents the thermal conductivity of graphene at 300 K; $T_{avg}$ represents the average temperature of the graphene membrane; α(T) is the thermal expansion coefficient of graphene at temperature T; $E_{Y}$ is Young’s modulus; and t, d, and ρ are the thickness, diameter, and mass density of graphene, respectively. Electrothermal tuning can avoid the capacitive softening effect in the electrostatic tuning method and can lead to an increase of the quality factor of graphene resonators.

### 4.2. Loss and Quality Factor

The loss is an important parameter in determining the performance of a resonator and is usually represented by quality factor Q. The expression for the quality factor is as follows:(2)$$\;Q=\frac{2\pi \Delta W}{W}=\frac{f_{0}}{\Delta f}$$
where the first term is the defining formula for the quality factor, which represents the ratio of the energy consumed per period to the total energy stored in the resonator; the second term is obtained by solving the standard damped equation, where $f_{0}$ is the resonant frequency and Δf is the frequency bandwidth. The *Q* value can be obtained by spectral measurements or time-domain measurement. It is worth mentioning that the time-domain measurement can effectively avoid the spectral spreading effect in spectral measurements [57]. Table 1 lists the quality factors of different graphene resonators from previous works.

A number of mechanisms can lead to the reduction of the quality factor. Imboden et al. summarized the losses in graphene resonators into the extrinsic and the intrinsic losses according to their source [59]. Extrinsic losses are introduced by system constraints or the external measurement environment. For example, the losses can be introduced by excitation and detection schemes, including ohmic loss introduced by electrical excitation [45,60] and magneto-dynamic loss [61] due to the magneto-dynamic excitation. Extrinsic losses include clamping loss [61], piezoelectric film loss [62], and so on, which are commonly found in mechanical resonators. Intrinsic losses are associated with the properties of graphene. The high-surface area-to-volume ratio of graphene makes the surface loss non-negligible. Surface loss is mainly caused by the presence of a large number of structural defects, impurities, and suspended bonds on the graphene surface, which play a dominant role at low temperatures and in the miniaturized resonators [63]. In addition, the hysteresis of stress and strain caused by interlayer friction and slip produces a unique loss of two-dimensional resonators. This explains that the quality factor of twisted and Bernal-stacked double-layer graphene resonators is obviously lower than that of single-layer resonators. The friction from the incommensurate twisted bilayer film is large enough to affect the loss of the 2D nano-mechanical resonator [64]. Another important factor for intrinsic loss is the thermoelastic loss. The temperature gradient of the vibrating membrane under the bending load leads to irreversible heat conduction and energy dissipation in the suspended devices [65]. In addition, experimental data shows that the quality factor of graphene resonators increases rapidly as the temperature decreases. Similar behavior also appears in other mechanical resonators based on two-dimensional materials [66,67]. Since the membrane tension increases at low temperature, the loss is reduced and the *Q* value increases accordingly [68], which implies that the tension plays an important role in determining the loss and the device performance of graphene resonators.

The source of non-linear damping is a complex problem. One explanation is that it mainly comes from two aspects: general loss path and geometric non-linearity [58]. For example, the non-linear damping from viscoelasticity can be derived by the single degree of freedom model with geometric non-linearity [69]. Non-linear-mode coupling is also one of the important sources of non-linear loss. Güttinger et al. studied the effect of non-linear coupling on the energy dissipation of the nano-electromechanical resonant system [70]. Through the energy decay test, the flow path of energy between different modes under high and low amplitudes is revealed. When the mode is decoupled at low amplitude, a very low energy decay rate can be observed. In order to minimize the non-linear damping and to obtain a large *Q* factor, Eichler et al. reduced the driving force until the motion became almost undetectable and they eventually obtained a graphene resonator with an ultra-high quality factor of 100,000 at 90 mK [58].

### 4.3. Non-Linear Effects

Due to the fact that graphene is an atomically thin membrane, the out-of-plane deflection of graphene is much larger than its thickness in a vibrating graphene sheet, which means that it is easy to trigger the non-linear effect in graphene resonators [71]. Such non-linearity will result in a small dynamic range. Dynamic range represents the linear operating range of a device, which is determined by the ratio between the upper limit of the signal to the non-linearity and the lowest detectable signal [72]. Different from one-dimensional mechanical resonators, the dynamic range of two-dimensional resonators (such as graphene) can be expressed as: $DR\propto 10\log\left(E_{Y}^{3/2}\rho_{3D}^{-1/2}rt\varepsilon^{5/2}\right)$ [72]. As a result, the initial strain ε of graphene can effectively increase the dynamic range. In addition, cancelling out partial non-linearities by adjusting the design parameters can also increase the dynamic range [73]. Parmar et al. increased the dynamic range of graphene resonators by 25 dB using these two methods [74].

Considering the non-linear effect, the Newtonian equation of the resonator can be expressed as follows [58]:
(3)$$\frac{md^{2}x}{dt^{2}}=-kx-\frac{\gamma dx}{dt}-\alpha x^{3}-\frac{\eta x^{2}dx}{dt}+F_{drive}\cos\left(2\pi ft\right)$$
where $\alpha x^{3}$ is the Duffing non-linear term and the positive or negative of the Duffing parameter α depends on the interaction between the non-linear term in the electrostatic force and the higher-order term of the mechanical restoring force [3,45]. If α is positive, then the resonance frequency increases as the amplitude of α increases. In contrast, if α is negative, then the frequency decreases as the amplitude increases. However, on the other hand, the existence of non-linear vibration is also beneficial to the partial application of graphene resonators. Dai et al. demonstrated that the existence of the non-linear effect can effectively improve the detection sensitivity of graphene resonators [71]. This work shows that the in-plane tension can play an important role in tunning the non-linearity of graphene resonance. The detection sensitivity of graphene resonators can be improved by using non-linear vibrations caused by geometric non-linear effects. In addition, the in-plane tension can also control the detection sensitivity of the graphene resonator, which operates with both harmonic and non-linear oscillation mechanisms. When graphene is in a highly non-linear vibration, the frequency shift caused by adsorption mass can be increased by six times.

## 5. Applications

### 5.1. Graphene Resonant Sensors

Graphene has excellent mechanical properties, such as ultra-high Young’s modulus and high fracture strength, as well as ultra-low surface density. In addition, the resonance properties of graphene are very sensitive to external physical parameters and the environment, such as force, attached mass, and acceleration. These characteristics make suspended graphene very suitable for various kinds of resonant sensors.

Resonant mass sensors are used to detect masses through the resonant frequency shift of graphene due to the attached mass. Based on the analysis methods of classical mechanics or nano-mechanics, a great deal of theoretical calculations and simulations of graphene resonant mass sensors are studied. For example, Dai et al. investigated the effect of non-linear vibration behavior and in-plane tension on the detection sensitivity of mass using the continuum elastic model [71]. Their results show that in-plane tension can play a key role in adjusting the non-linearity of graphene resonance. It has been found that the detection sensitivity of graphene resonators can be improved by using non-linear vibrations. Natsuki et al. studied the detection performance of bilayer graphene in nanoscale mass sensors and found that bilayer graphene provides higher detection sensitivity compared to monolayer graphene [75]. Furthermore, plate theory based on non-local continuum theory was also used to investigate graphene resonant mass sensors [76,77]. Furthermore, Karličić et al. introduced a magnetic field into the theoretical model for improving the sensitivity [77]. The increase in the magnetic field will increase the frequency shift of the resonance sensor, thereby improving its detection sensitivity. On the other hand, considering the damping characteristics of the sensor system, it has been shown that it reduces the frequency shift, thereby reducing the sensitivity characteristics of the device. Molecular dynamics simulations in nano-mechanics are commonly used to study the resonant properties of graphene. Xiao et al. proposed to use the first three resonance modes of the mass sensor to compensate the frequency shift caused by stress [78]. The simulation results show that increasing the prestress in the stretched graphene can significantly improve the sensitivity, although it is difficult to accurately determine the quality through the stress-dependent fundamental frequency shift. The attached mass in the middle of the graphene sheet significantly reduces the resonance frequency of mode 11 and the effect on bot mode 21 and mode 22 is negligible, as shown in Figure 6a.

The pressure sensors based on graphene nano-mechanical resonators have been studied by finite element simulation [79]. The simulation results indicate a linear variation of the resonant frequency with pressure. When graphene is monolayer, the pressure sensitivity can reach 26,838 Hz/kPa, which is two orders of magnitude higher than that of the conventional resonant pressure sensor. Dolleman et al. firstly fabricated a graphene resonant pressure sensor, as shown in Figure 7a [34]. There is a pressure difference between the gas sealed by the graphene membrane and the external gas to ensure that the graphene stiffness is changed, resulting in a resonant frequency shift. The experimental results show that the measurement response is 9000 Hz/mbar, which is 45 times higher than that of the most advanced squeeze film pressure sensor based on MEMS.

Yet, the exploration of graphene resonant accelerometers is at the preliminary stage. Kang et al. studied the resonant accelerometers based on graphene nano-ribbons [80] and crossroad-type graphene [81] through molecular dynamics simulations. The simulation results show that the resonant frequency increases linearly with acceleration on the logarithmic–logarithmic scale. Furthermore, Byun et al. investigated a graphene resonant accelerometer with attached mass by molecular dynamics simulations as well [82]. As the attached mass increases, the sensitivity decreases, while the sensing range remains the same. When the reference frequency is defined as the limit of the sensing range, acceleration can be sensed by fitting a function regardless of the additional mass and then the acceleration can be detected with a very high sensitivity. Shi et al. designed a differential graphene resonant accelerometer, as shown in Figure 6c [83]. By using the finite element simulation calculation, the sensitivity of the accelerometer can reach 21,224 Hz/g in the acceleration range of 0–1000 g, achieving highly sensitive acceleration measurements. Recently, Fan et al. have made a breakthrough in the preparation of a graphene resonant accelerometer, realizing suspended graphene ribbons with suspended masses, which are able to measure accelerations of the order of tens of micro-g [41]. These sensors are manufactured using processes compatible with large-scale semiconductor manufacturing technologies to produce NEMS accelerometers with a chip area that is at least two orders of magnitude smaller than traditional state-of-the-art silicon accelerometers.

### 5.2. Other Applications

In addition to measuring and sensing the external parameters, the graphene resonator can also be used to measure the intrinsic properties of graphene. Guan et al. fabricated a graphene-based strain sensor, in which the strain is adjusted by bending the substrate [14]. The strain of graphene can be obtained by analyzing the relationship between non-linear dynamics and gate voltage. The results show that the resonant frequency is significantly influenced by the applied strain and is related to the size of graphene. Since the resonant frequency is sensitive to the temperature, the thermal properties of graphene also can be studied in a resonance scheme as well. Singh et al. measured the thermal expansion coefficient of monolayer graphene from 30 K to 300 K using a graphene resonator [29]. The result shows that the thermal expansion of graphene is negative for all temperatures between 300 K and 30 K. As the temperature decreases, the electromechanical mode of positive dispersion evolves into the electromechanical mode of negative dispersion. Ye et al. measured the thermal expansion coefficient and electrical conductivity of graphene from 300 K to 1200 K by electrothermal tuning [56]. The thermal expansion coefficient and thermal conductivity of these devices can be extracted using the measured frequency and temperature changes. Graphene’s unique negative thermal expansion coefficient and its excellent resistance to high temperatures can be used to design highly adjustable and robust graphene transducers for harsh and extreme environments [56].

Graphene resonators can also provide a feasible scheme for detecting fundamental physical properties. For example, Chen et al. controlled the electron energy in the Landau level under an external magnetic field through the graphene resonators. In this way, the energy gap between the adjacent Landau levels and the energy conversion between the energy gaps are measured [84]. In addition, graphene resonators can be served as a single-electron transistor. Luo et al. found that there is a strong coupling between its mechanical motion and single-electron tunneling, and the reaction force caused by the fluctuation of electron transport leads to the frequency shift exceeding 100 KHz, which is tens of times larger than its minimum linewidth [85].

Recently, graphe0ne drum resonators have been used for pneumatic testing due to the flexibility and gas tightness of graphene membranes [86]. As shown in Figure 7a, Davidovikj et al. coupled two graphene drums through a trench and designed local electrodes to drive the two graphene drums separately, which enables an electrical control and manipulation of the gas flow inside the channel between these two drums [87]. The displacement of each drum can be measured at atmospheric pressure as a function of the frequency of the electrostatic driving force, which provides a proof-of-principle of graphene gas pumps at the nanoscale. Rosłoń et al. pumped gas sealed by the graphene membrane into nanopores by using photothermal excitation, as shown in Figure 7b [88]. By detecting the time delay between the driving force and the motion of the graphene membrane, the permeation time constants of different gases through the nanopores can be obtained, providing a nano-mechanical method to calculate the molecular mass of the gas.

The high tensile strength and mechanical tunability of 2D materials such as graphene have attracted more and more attention to 2D phononic crystals in recent years. The feasibility of phononic crystals based on graphene electromechanical resonators is studied theoretically and experimentally [89,90]. In the experiment, Zhang et al. fabricated a graphene-based nano-electromechanical periodic array with a quasi-continuous spectrum [91]. The spectrum had a range of hundreds MHz and was regulated by gate voltage. The appearance of banded frequency, which was caused by the hybridization of mechanical modes, proved the formation of graphene phononic crystals. In addition, Kirchhof et al. studied the local defect modes in graphene phononic crystals [92]. They fabricated a tunable monolayer graphene phononic crystal and localized a defect mode in the megahertz bandgap. With the modulation of induced pressure, the tuning range of the defect mode up to more than 350% was realized, which brings new prospects for strain engineering.

## 6. Other Two-Dimensional Materials

### 6.1. Graphene-Based Heterojunction Resonators

The Van der Waals heterostructure is a unique structure in two-dimensional materials. By stacking different types of two-dimensional (2D) structures, the mechanical, electrical, and thermal properties of heterojunctions can be actively regulated, which brings an endless potential of two-dimensional resonators.

Inoue et al. fabricated graphene-molybdenum disulfide heterojunction resonators [93]. By stacking the two materials with opposite thermal expansion coefficients in the vertical direction, the thermal expansion coefficient of the heterojunction is reduced to 1/3 of the monolayer graphene and the effect of temperature on the resonance frequency is reduced. In order to reduce the electrical loss, Will et al. fabricated a graphene-niobium diselenide-graphene heterojunction resonator [94]. The strong conductivity of niobium diselenide makes the resonator have lower electrical loss. At low temperature, the quality factor can reach 245,000, which is rare in graphene resonators. Their work provides a new idea for the development of nano-mechanical resonators with high quality and low loss. In addition to improving the performance of two-dimensional resonators, heterojunction resonators can also be used to study the interaction between heterojunction layers. Kim et al. fabricated heterojunction resonators of monolayer molybdenum disulfide and monolayer graphene. It is found that the asymmetric tuning curve is caused by interlayer slip or fold [95].

### 6.2. Other Two-Dimensional Materials-Based Resonators

With the continuous discovery of new two-dimensional materials, some of them have become new candidates for nano-electromechanical resonators, such as transition metal dichalcogenides (TMD) [57,66,96], hexagonal boron nitride (h-BN) [97], black phosphorus (BP) [98], and so on. In addition to the common characteristics of two-dimensional electromechanical resonators, different two-dimensional materials also show their unique properties, which expands the scope of research and application of nano-electromechanical resonators. For example, TMD are widely used in the preparation of nano-mechanical resonators, except for graphene. Several layers of TMD are indirect bandgap semiconductors, while monolayer TMD are direct bandgap semiconductors, and the bandgap can be adjusted by strain; this makes up for the lack of bandgap in graphene and brings a bright prospect for nano-electromechanical systems with mechanically adjustable optoelectronic properties.

## 7. Conclusions

Graphene is the first two-dimensional material to be discovered. It has atomic thickness, an extremely high Young’s modulus and fracture strength, and very low surface density, which make graphene an ideal material for the fabrication of nano-electromechanical resonators. The large surface-to-volume ratio induced by ultra-thin thickness also makes graphene resonators have great potential in the application of pressure, mass, acceleration, and other ultra-sensitive detectors. Since the first fabrication of graphene resonator in 2007, after more than ten years of development, great breakthroughs have been made in the fabrication technology and weak signal detection scheme, and the theory on the resonance characteristic also has been constantly updated. The application of graphene resonators is gradually moving from simulation to experiment and from the classical field to quantum field.

However, there are still many challenging issues in graphene resonators to be solved in this stage. We summarize these challenges and the main directions of future research as follows.

Fabrication of high-quality graphene

The repeatability of graphene preparation is too weak due to the uncontrollable factors such as graphene strain, doping, defects, and so on. Additionally, the fabrication of graphene with large-scale, high-quality, uniform, and controllable thickness is still a challenge.

Strain engineering

The strain has a strong influence on the resonant performance of graphene resonators. In the process of fabrication, graphene is easy to produce residual strain, which is usually harmful. Thus, the approaches to actively control the strain of graphene is important.

Device integration

After successful fabrication of graphene resonators, it is still a challenge to integrate graphene resonators into electronic systems and necessary to establish a set of integrated processes with a high efficiency, low cost, and high success rate.

Loss source of graphene resonators

The loss of graphene resonators is very diverse but the source and mechanism of many losses are not clear; in particular, the relationship between loss and temperature still has a lack of accurate description. Therefore, it is necessary and urgent to study the loss source of graphene.

Research of non-linear effects

Since the graphene resonator has a high level of non-linearity, resonant sensors based on the non-linear behavior are also widely concerning. In particular, the non-linear vibration has been proved to improve the sensitivity of sensors, which attracts attention to application in the non-linear regime.

The expansion of the application field

It is very attractive to further expand the application field of graphene resonators from the classical field to quantum field. Considering the atomic thickness of graphene, the graphene-based resonator can reach an extremely high frequency, small mass, and dramatical large zero-point motion, which makes it an ideal device for quantum-electromechanical systems.

In general, graphene resonators promote the development of micro-nano-electromechanical systems to miniaturization, ultra-low power consumption, and ultra-high sensitivity. With the continuous progress of the device fabrication and detection technique, and with the improvement of the theoretical model, graphene resonators will certainly open the potential for wide applications in various high-tech fields.

## Figures and Tables

**Figure 1 micromachines-13-00215-f001:**
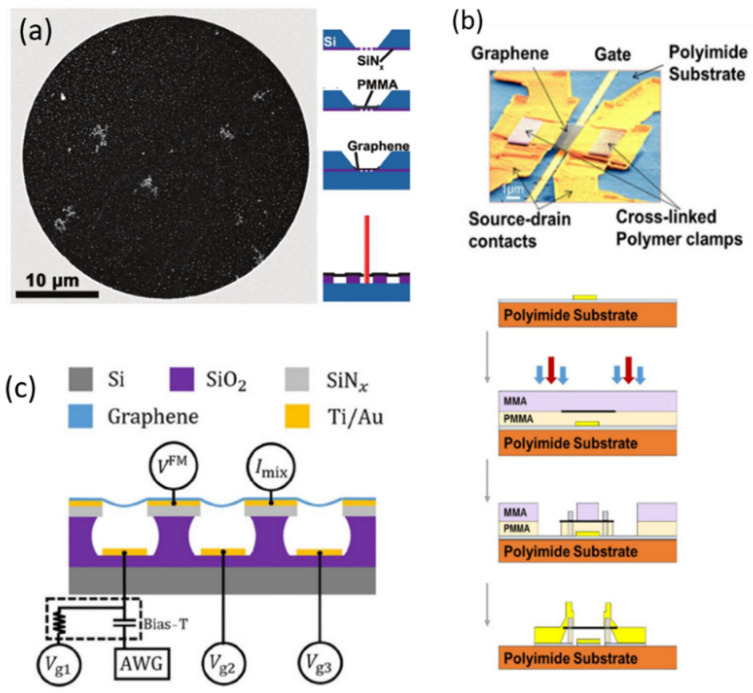
Fabrication of graphene resonators. (**a**) Transfer the CVD graphene to suspended nitride film with prefabricated holes [13]. (**b**)Transfer the methyl methacrylate (MMA) film with graphene to the polymethyl methacrylate (PMMA)-covered substrate followed by non-etching suspension process [14]. (**c**) Simultaneous deposition of source-drain electrodes and local gate electrodes after etching [10].

**Figure 2 micromachines-13-00215-f002:**
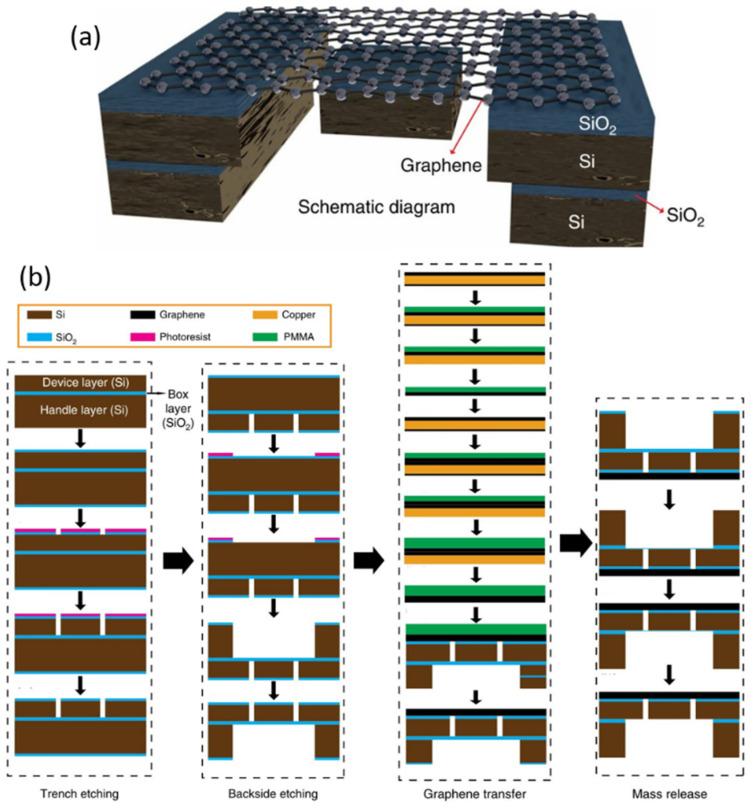
Graphene membranes with suspended silicon proof masses [30]. (**a**) The structure of device. (**b**) The fabrication process of device.

**Figure 3 micromachines-13-00215-f003:**
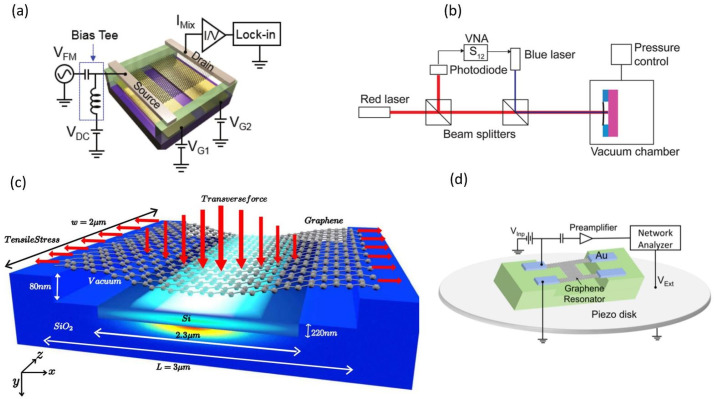
Excitation schemes for graphene resonators. (**a**) Electrostatic excitation scheme with a bipolar region [32]. (**b**) Optic excitation scheme with intensity-modulated laser [34]. (**c**) Excitation scheme with optical gradient forces [36]. (**d**) Piezoelectric excitation scheme by placing graphene on a piezo disk [19]. As a zero-bandgap semimetal, the conductance of graphene depends on the gate capacitance and gate voltage. Therefore, the mechanical motion of graphene can be reflected in the output current as the gate voltage remains constant [18]. However, in practical measurements, it is difficult to measure the output currents directly due to the large parasitic capacitance of the system. Thus, the mixing technique is often used to read out the signal converted into low frequency without losing information about amplitude [42], which involves replacing the RF signal with an amplitude modulation (AM) or frequency modulation (FM) signal applied to the source or drain. Then, a lock-in amplifier was used to read out the current at the mixing frequency, as shown in Figure 4a. One of the advantages of the FM technique over the AM route is that there is no electrical background in the mixed currents [43].

**Figure 4 micromachines-13-00215-f004:**
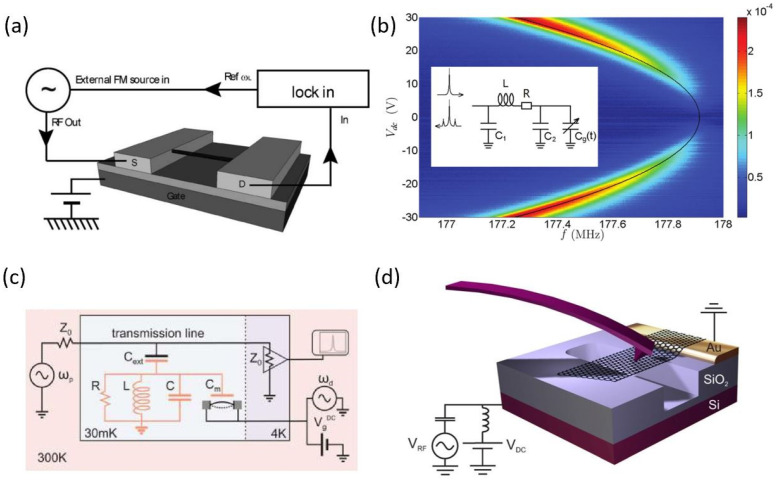
Detection schemes for graphene resonators. (**a**) Mixing technique with a frequency modulation (FM) signal [42]. (**b**) The RF cavity readout scheme with coupling of the graphene resonator to an LC circuit [45]. (**c**) Detection scheme with coupling of a graphene resonator to a superconducting microwave cavity [23]. (**d**) Imaging the vibration eigenmodes of graphene with SFM [48].

**Figure 5 micromachines-13-00215-f005:**
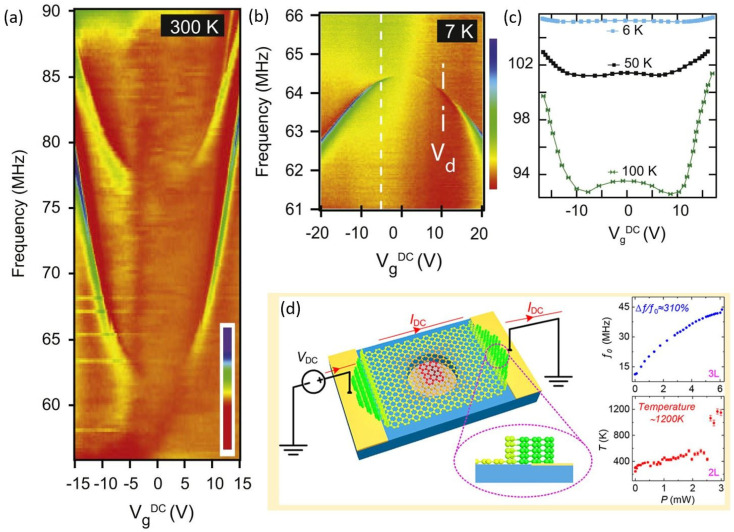
Frequency tuning of graphene resonator. (**a**) Spring hardening effect [29]. (**b**) Spring softening effect [29]. (**c**) The competition between the two effects with ‘W’ tuning behavior [29]. (**d**) Frequency tuning by Joule heating [56].

**Figure 6 micromachines-13-00215-f006:**
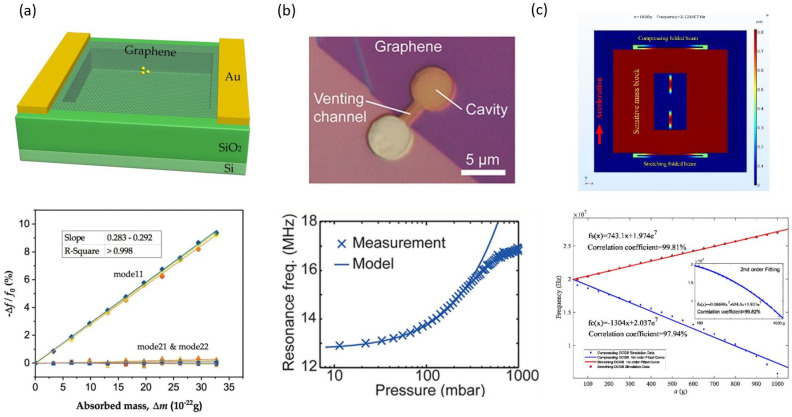
Graphene resonant sensors. (**a**) A graphene resonant mass sensor with simulation results [78]. (**b**) A graphene resonant pressure sensor with measurement results [34]. (**c**) A graphene resonant accelerator with simulation results [83].

**Figure 7 micromachines-13-00215-f007:**
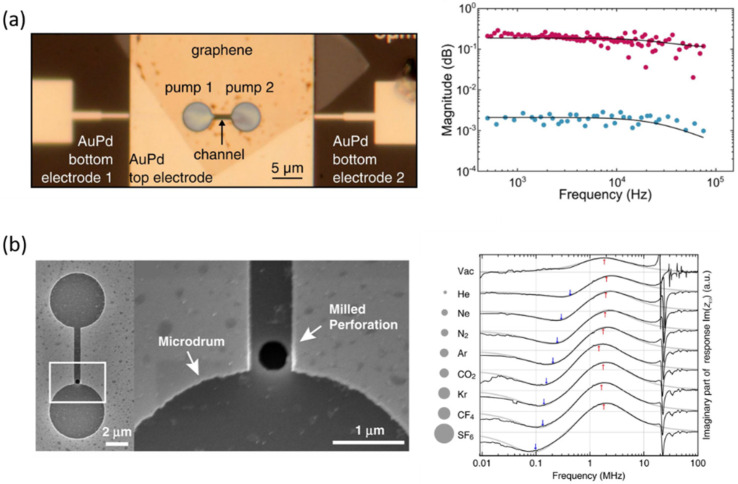
Graphene drum resonators for pneumatic testing. (**a**) Graphene gas pumps [87] and (**b**) graphene drum with a milled nanopore [88].

**Table 1 micromachines-13-00215-t001:** The quality factors of different graphene resonators.

Device Image	Quality Factor	Temperature	Year	Reference
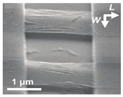	9000	10 K	2010	[18]
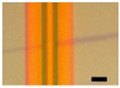	10,000	77 K	2010	[22]
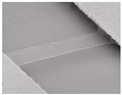	100,000	100 mK	2011	[58]
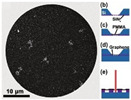	2400	300 K	2011	[13]
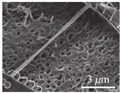	7000	300 K	2012	[24]
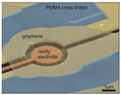	100,000	4.2 K	2014	[23]
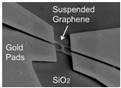	1180	300 K	2015	[19]
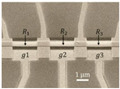	111,000	10 mK	2020	[10]

## Data Availability

The data are included in the main text.

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
