# Peer review of "A Review on Graphene-Based Nano-Electromechanical Resonators: Fabrication, Performance, and Applications"

_micromachines, 2022, doi:10.3390/mi13020215_

Round 1

Reviewer 1 Report

This review is good, interesting and useful.

Small comments.

  1. For all variables formula (1) does not have a complete description.
    2. The conductivity of graphene depends on the frequency. What frequency values were taken into account for the calculation of the quality factor?

Author Response

Dear Editors and Reviewers: 

Thank you for the comments concerning our manuscript entitled “A review on graphene-based nano-electromechanical resonators: fabrication, performance, and applications” (ID: micromachines-1556907). Those comments are all valuable and very helpful for revision and improvement of our paper, as well as the next research. The main corrections for the paper and the responds to the reviewer’s comments are as follows:

(The position of the corresponding modification in the manuscript is based on "unmarked" mode)

Response to reviewer #1:

Q1. For all variables formula (1) does not have a complete description.

Response: We have re-written this part according to the your suggestion.

We have now modified the following sentences:

Page 3, Line 271-274:

The tuning formula can be expressed as follow:

where  represents the thermal conductivity of graphene at 300 K,  represents the average temperature of the graphene membrane,  is the thermal expansion coefficient of graphene at temperature ,  is Young’s modulus, ,  and  are the thickness, diameter and mass density of graphene, respectively.

Q2. The conductivity of graphene depends on the frequency. What frequency values were taken into account for the calculation of the quality factor?

Response: The conductivity of graphene can be expressed as,

where  and  are the real and imaginary parts of the conductivity,  is the prefactor known as the Drude weight, and  is the electron scattering rate of graphene. From the above formula that, when , the conductivity is independent of frequency. The  of exfoliation or chemical vapor deposition (CVD) graphene is . For graphene resonators, the working frequency is usually dozens of MHz, so graphene’s conductivity should be frequency-independent and approximately equal to the DC conductivity (2D Mater. 3, 015010 (2016)).

Other language problems have been checked and modified directly in the manuscript.

Reviewer 2 Report

In the submitted manuscript, Xiao et al. provides a review on the fabrication, performance, and applications of graphene-based nanoelectromechanical resonators. As a frontier topic in the field of nanoscience and nanotechnology, graphene-based nanoelectromechanical resonators attracted significant research interests. The topic is well qualified for this special issue. Overall, this review is up-to-date, and well organized. However, I have several comments to the authors before recommending publication:

  • Fabrication method: In lines 93-96, the authors claim that ‘In the first process, the graphene is finally placed on the target substrate, so the graphene will not be exposed to polymer reagents and has high quality and integrity’. However, for most practical ways for transferring graphene onto pre-patterned substrates, direct contacting between graphene and polymers are inevitable. Also, using PMMA as transfer agent, PMMA can be overexposed thus fix the membrane onto the substrate [see Ref. 30].
  • Excitation and detection: In both electrical and optical excitation schemes, besides direct driving on resonance, parametric driving is widely applied, such as [Nat. Nanotechnol. 11, 747–751 (2016)][Nat. Nanotechnol. 11, 741–746 (2016)][Nat. Commun. 12, 1099 (2021)].
  • Loss mechanism: In section 4.2, I think the authors need to mention nonlinear effects here for the completeness of the structure. Also, nonlinear coupling between different modes is an important source for the devices, see [Nat. Nanotechnol. 12, 631-636 (2017)]. In addition, recently interlayer interactions induced dissipations are recognized as another important source, which is special for 2D resonators, see [Nano Lett. 19, 8058-8065 (2021)].
  • Applications: Graphene resonators are also sensitive detectors for fundamental physical properties, such as Landau Level [Nat. Phys. 12, 240-244 (2016)], single-electron tunneling, etc. Besides, with a more complex architecture, graphene nanomechanical resonator array is investigated to form phononic crystal, both theoretically and experimentally, see [Phys. Rev. Appl. 11, 024024 (2019)][Phys. Rev. Appl. 15,034015 (2021)][Nano Lett. 5, 2174–2182 (2021)][Nano Lett. 20, 8571-8578 (2021)]
  • Besides single material, 2D heterostructures consisting of graphene and other 2D materials are used to fabricate nanomechanical resonators. I think the authors need to mention this.
  • Typo: In line 116, it should be ‘Ti and Cr’, rather than ‘Ti and Cd’.

Author Response

Dear Editors and Reviewers: 

Thank you for the comments concerning our manuscript entitled “A review on graphene-based nano-electromechanical resonators: fabrication, performance, and applications” (ID: micromachines-1556907). Those comments are all valuable and very helpful for revision and improvement of our paper, as well as the next research. The main corrections for the paper and the responds to the reviewer’s comments are as follows:

(The position of the corresponding modification in the manuscript is based on "unmarked" mode)

Response to reviewer #2:

Q1. Fabrication method: In lines 93-96, the authors claim that ‘In the first process, the graphene is finally placed on the target substrate, so the graphene will not be exposed to polymer reagents and has high quality and integrity’. However, for most practical ways for transferring graphene onto pre-patterned substrates, direct contacting between graphene and polymers are inevitable. Also, using PMMA as transfer agent, PMMA can be overexposed thus fix the membrane onto the substrate [see Ref. 30].

Response: It is really true as you pointed that most of the transferring ways will bring graphene into contact with polymers such as PMMA, PVA, PDMS. So, we have re-written this part according to your suggestion. In addition, we have added an introduction to the fabrication method of overexposing PMMA polymer to fix graphene membrane as you mentioned in the manuscript.

We have now modified the following sentences:

Page 2, Line 87-90:

In the first process, the graphene is finally placed on the target substrate, so the graphene will not be exposed to polymer resists without etching or evaporating the electrodes and has higher quality and integrity [22].

Page 2, Line 90-92:

But in this case, the graphene is not held by metal electrodes or other structures and thus remains unstable. One alternative is to use PMMA as a transfer agent to locate graphene to the target substrate, and then overexpose the PMMA to form graphene clamping [23].

Q2. Excitation and detection: In both electrical and optical excitation schemes, besides direct driving on resonance, parametric driving is widely applied, such as [Nat. Nanotechnol. 11, 747–751 (2016)][Nat. Nanotechnol. 11, 741–746 (2016)][Nat. Commun. 12, 1099 (2021)].

Response: We have recognized the importance of parametric driving and add a corresponding description in the manuscript.

We have now modified the following sentences:

Page 5, Line 185-192:

 In addition to direct excitation, parametric excitation is also widely used in electrical or op-tical excitation schemes [39,40], which plays a positive role in improving the quality factor, signal-to-noise ratio and sensitivity of graphene resonators. Dolleman et al. periodically modulated the tension and stiffness of graphene monolayer through photothermal effect, thus parametrically excited multiple resonance modes and realized parametric resonance and parametric amplification of graphene resonators [41]. In the all-electric scheme, Mathew et al realized the dynamic modulation of the multi-mode coupling of the graphene resonator through parametric excitation, which provided the feasibility for the quantum mechanical experiments at low temperature [42].

Q3. Loss mechanism: In section 4.2, I think the authors need to mention nonlinear effects here for the completeness of the structure. Also, nonlinear coupling between different modes is an important source for the devices, see [Nat. Nanotechnol. 12, 631-636 (2017)].

Response: We have recognized the importance of nonlinear effects in the loss mechanism and add a corresponding description in the manuscript.

We have now modified the following sentences:

Page 8, Line 314-324:

The source of nonlinear damping is a complex problem. One explanation is that it mainly comes from two aspects: general loss path and geometric nonlinearity [62]. For example, the non-linear damping from viscoelasticity can be derived by single degree of freedom model with geo-metric nonlinearity [73]. Nonlinear mode coupling is also one of the important sources of nonlinear loss. Güttinger et al. studied the effect of nonlinear coupling on the energy dissipation of nano-electromechanical resonant system [74]. Through the energy decay test, the flow path of energy between different modes under high and low amplitudes is revealed. When the mode is decoupled at low amplitude, a very low energy decay rate can be observed. In order to minimize the nonlinear damping and to obtain a large Q factor, Eichler et al. reduced the driving force until the motion becomes almost undetectable, and they eventually obtained a graphene resonator with an ultra-high quality factor of 100000 at 90mK [62].

Q4. In addition, recently interlayer interactions induced dissipations are recognized as another important source, which is special for 2D resonators, see [Nano Lett. 19, 8058-8065 (2021)].

Response: We have recognized the importance of interlayer interactions induced dissipations and add a corresponding description in the manuscript.

We have now modified the following sentences:

Page 8, Line 301-305:

In addition, the hysteresis of stress and strain caused by interlayer friction and slip produces a unique loss of two-dimensional resonators. This explains that the quality factor of twisted and Bernal-stacked double-layer graphene resonators is obviously lower than that of single-layer. The friction from the incommensurate twisted bilayer film is large enough to affect the loss of the 2D nano-mechanical resonator [68].

Q5. Applications: Graphene resonators are also sensitive detectors for fundamental physical properties, such as Landau Level [Nat. Phys. 12, 240-244 (2016)], single-electron tunneling, etc.

Response: We have recognized that graphene resonators have an important application in the detection for fundamental physical properties and added a corresponding description in the manuscript.

We have now modified the following sentences:

Page 11, Line 430-437:

Graphene resonators can also provide a feasible scheme for detecting fundamental physical properties. For example, Chen et al. control the electron energy in the Landau level under an external magnetic field through the graphene resonators. In this way, the energy gap between the adjacent Landau levels and the energy conversion between the energy gaps are measured [88]. In addition, graphene resonator can be served as a single-electron transistor. Luo et al. found that there is a strong coupling between its mechanical motion and single-electron tunneling, and the reaction force caused by the fluctuation of electron transport leads to the frequency shift exceeding , which is tens of times larger than its minimum linewidth [89].

Q6. Besides, with a more complex architecture, graphene nanomechanical resonator array is investigated to form phononic crystal, both theoretically and experimentally, see [Phys. Rev. Appl. 11, 024024 (2019)][Phys. Rev. Appl. 15,034015 (2021)][Nano Lett. 5, 2174–2182 (2021)][Nano Lett. 20, 8571-8578 (2021)].

Response: We have recognize the importance of graphene-based phononic crystal and add a corresponding description in the manuscript.

We have now modified the following sentences:

Page 11, Line 449-459:

The high tensile strength and mechanical tunability of 2D materials such as graphene have attracted more and more attention to 2D phononic crystals in recent years. The feasibility of pho-nonic crystals based on graphene electromechanical resonators is studied theoretically and ex-perimentally [93,94]. In the experiment, Zhang et al. fabricated a graphene-based nano-electromechanical periodic array with quasi-continuous spectrum [95]. The spectrum has a range of hundreds MHz and is regulated by gate voltage. The appearance of banded frequency which caused by the hybridization of mechanical modes proves the formation of graphene phononic crystals. In addition, Kirchhof et al. studied the local defect modes in graphene phononic crystals [96]. They fabricated a tunable monolayer graphene phononic crystal and localized a defect mode in the megahertz bandgap. With the modulation of induced pressure, the tuning range of defect mode up to more than 350% is realized, which brings new prospects for strain engineering.

Q7. Besides single material, 2D heterostructures consisting of graphene and other 2D materials are used to fabricate nanomechanical resonators. I think the authors need to mention this.

Response: We have add a corresponding description in the manuscript.

We have now modified the following sentences:

Page 12, Line 465-481:

Van der Waals heterostructure is a unique structure in two-dimensional materials. By stacking different types of two-dimensional (2D) structures, the mechanical, electrical and thermal properties of heterojunctions can be actively regulated, which brings endless potential of two-dimensional resonators.

Inoue et al. have fabricated graphene-molybdenum disulfide heterojunction resonators [97]. By stacking the two materials with opposite thermal expansion coefficients in the vertical direction, the thermal expansion coefficient of the heterojunction is reduced to 1/3 of the monolayer graphene, and the effect of temperature on the resonance frequency is reduced. In order to reduce the electrical loss, Will et al. fabricated a graphene-niobium diselenide-graphene heterojunction resonator [98]. The strong conductivity of niobium diselenide makes the resonator have lower electrical loss. At low temperature, the quality factor can reach 245000, which is rare in graphene resonators. Their work provides a new idea for the development of nano-mechanical resonators with high quality and low loss. In addition to improving the performance of two-dimensional resonators, heterojunction resonators can also be used to study the interaction between heterojunction layers. Kim et al. have fabricated heterojunction resonators of monolayer molybdenum disulfide and monolayer graphene. It is found that the asymmetric tuning curve is caused by interlayer slip or fold [99].

Q8. Typo: In line 116, it should be ‘Ti and Cr’, rather than ‘Ti and Cd’.

Response: We are very sorry for our incorrect writing. We have re-written this word in the manuscript.

We have now modified the following sentences:

Page 3, Line 110-111:

Ti [11] or Cr [19] is usually deposited in the middle of graphene and Au as an adhesive layer.

Other language problems have been checked and modified directly in the manuscript.

Reviewer 3 Report

  1. Please add the future directions of Resonators
  2. what are the issues in fabrication of graphene based resonators
  3. Give the elaborative  conclusions and remarks of the graphene based resonators
  4. why only Graphene resonators are important than other 2D materials like MOS2, WS2 etc

Author Response

Dear Editors and Reviewers: 

Thank you for the comments concerning our manuscript entitled “A review on graphene-based nano-electromechanical resonators: fabrication, performance, and applications” (ID: micromachines-1556907). Those comments are all valuable and very helpful for revision and improvement of our paper, as well as the next research. The main corrections for the paper and the responds to the reviewer’s comments are as follows:

(The position of the corresponding modification in the manuscript is based on "unmarked" mode)

Response to reviewer #3:

Q1. Please add the future directions of Resonators.

Response: We speculate that the main development direction of graphene resonator in the future may include the following three areas: 1. Loss source of graphene resonator; 2. Research and application of nonlinear effects; 3. The expansion of graphene resonator at the application stage.

  1. Loss source of graphene resonator

High loss and low quality factor of graphene resonators at room temperature are severe   challenges that hinder their practical application. The loss of graphene resonators is very diverse, but the source and mechanism of many losses are not clear, especially the relationship between loss and temperature is still lack of accurate description. Therefore, it is necessary and urgent to study the loss source of graphene.

  1. Research and application of nonlinear effects

The graphene resonator has a high level of nonlinearity, and even when driven only by thermal noise, the graphene resonator may operate within a nonlinear regime. Therefore, in order to make the graphene resonator operate in the linear regime, it is usually necessary to have as little driving force as possible and to work at low temperature. In addition, since the linear dynamic range of nano-mechanical resonators is very limited, resonant sensors based on the nonlinear behavior are also widely concerned. In theory, the nonlinear vibration has been proved to improve the sensitivity of graphene mass sensor. Therefore, the application in nonlinear regime has attracted a lot of attention.

  1. The expansion of application field

It is very attractive to further expand the application field of graphene resonators from classical field to quantum field. Since the atomic thickness of graphene, the graphene-based resonator can reach extremely high frequency, small mass and dramatical large zero-point motion, which makes it an ideal device for quantum-electromechanical systems.

We will state the modification in the manuscript in our reply to question 3.

Q2. What are the issues in fabrication of graphene based resonators.

Response: We think that the main issues in fabrication of graphene resonators include the following three points at the present stage:

1.Control of prestress

Since the bending stiffness of graphene is very small, the strain has a strong influence on the resonant frequency, nonlinear effect and quality factor of graphene resonators. In the process of fabrication, due to the adhesion of graphene to the channel wall and other reasons, graphene is easy to produce residual strain which is usually harmful. However, the approaches to actively control the pre-strain of graphene is to be explored.

  1. Fabrication of large-scale and high-quality graphene

Although graphene resonators with high quality factor have been fabricated in the experiment, the repeatability of graphene preparation is too weak due to the uncontrollable factors such as graphene strain, doping, defects and so on. And the fabrication of graphene with large scale, high quality, uniform and controllable thickness is still a challenge, which seriously affects the popularization of graphene resonator in engineering application.

  1. Device integration

Although there have been many successful cases of fabrication of graphene resonators, it is still a challenge to integrate graphene resonators into electronic systems and control costs. Therefore, when the two-dimensional material is applied to the nano-electromechanical system, it is necessary to establish a set of integrated process with high efficiency, low cost and high success rate.

We will state the modification in the manuscript in our reply to question 3.

Q3. Give the elaborative conclusions and remarks of the graphene based resonators.

Response: We expand the conclusion, comment on the state-of-the-art of graphene resonators, and list the main challenges at present and the development direction in the future.

We have now modified the following sentences:

Page 13, Line 496-536:

Graphene is the first two-dimensional material to be discovered. It has atomic thickness, extremely high Young's modulus and fracture strength, and very low surface density, which make graphene an ideal material for the fabrication of nano-electromechanical resonators. The large surface-to-volume ratio induced by ultra-thin thickness also makes graphene resonators have great potential in the application of pressure, mass, acceleration and other ultra-sensitive detectors. Since the first fabrication of graphene resonator in 2007, after more than ten years of development, great breakthroughs have been made in the fabrication technology and weak signal detection scheme, and the theory on resonance characteristic also has been constantly updated. The application of graphene resonator is gradually moving from simulation to experiment, and from classical field to quantum field.

However, there are still many challenging issues in graphene resonators to be solved in this stage. We summarize these challenges and the main directions of future research as follows.

  • Fabrication of high-quality graphene

The repeatability of graphene preparation is too weak due to the uncontrollable factors such as graphene strain, doping, defects and so on. And the fabrication of graphene with large scale, high quality, uniform and controllable thickness is still a challenge.

  • Strain engineering

The strain has a strong influence on the resonant performance of graphene resonators. In the process of fabrication, graphene is easy to produce residual strain which is usually harmful. Thus, the approaches to actively control the strain of graphene is important.

  • Device integration

After successful fabrication of graphene resonators, it is still a challenge to integrate graphene resonators into electronic systems and necessary to establish a set of integrated process with high efficiency, low cost and high success rate.

  • Loss source of graphene resonator

The loss of graphene resonators is very diverse, but the source and mechanism of many losses are not clear, especially the relationship between loss and temperature is still lack of accurate description. Therefore, it is necessary and urgent to study the loss source of graphene.

  • Research of nonlinear effects

Since the graphene resonator has a high level of nonlinearity, resonant sensors based on the nonlinear behavior are also widely concerned. Especially, the nonlinear vibration has been proved to improve the sensitivity of sensor, which attracts attention of the application in nonlinear regime.

  • The expansion of application field

It is very attractive to further expand the application field of graphene resonators from classical field to quantum field. Since the atomic thickness of graphene, the graphene-based resonator can reach extremely high frequency, small mass and dramatical large zero-point motion, which makes it an ideal device for quantum-electromechanical systems.

In general, graphene resonators promote the development of micro-nano electromechanical systems to miniaturization, ultra-low power consumption and ultra-high sensitivity. With the continuous progress of the device fabrication and detection technique and the improvement of the theoretical model, graphene resonators will certainly open the potential for widely application in various high-tech fields.

Q4. Why only Graphene resonators are important than other 2D materials like MOS2, WS2 etc.

Response: Due to the transition from bulk phase to a few layers to monolayer limits, two-dimensional materials have unique mechanical, optical and electrical properties. For example, many two-dimensional materials show high Young's modulus, carrier mobility, surface area-volume ratio, low surface density and so on. Therefore, the nano-mechanical resonators based on two-dimensional materials have a lot in common in fabrication, excitation and resonance performance. As a two-dimensional material with the longest research history and the widest range of applications, graphene has become a representative material for the study of nano-mechanical resonators in this paper.

       In addition to the above similarities, other two-dimensional materials also show their unique properties. For example, monolayer transition metal dichalcogenides (TMD) are direct band gap semiconductors, which make up for the lack of band gap in graphene and have the potential to fabricate nanoelectromechanical systems with mechanically adjustable optoelectronic properties. Therefore, we will add a brief description of the other two-dimensional materials and graphene-based heterojunction resonators to enrich the structure of the article.

We have now modified the following sentences:

Page 12, Line 483-493:

With the continuous discovery of new two-dimensional materials, some of them have be-come new candidates for nano-electromechanical resonators, such as transition metal dichalco-genides (TMD) [100-102], hexagonal boron nitride (h-BN) [103], black phosphorus (BP) [104] and so on. In addition to the common characteristics of two-dimensional electromechanical resonators, different two-dimensional materials also show their unique properties, which expands the scope of research and application of nano-electromechanical resonators. For example, TMD are widely used in the preparation of nano-mechanical resonators except graphene. Several layers of TMD are indirect bandgap semiconductors, while monolayer TMD are direct bandgap semiconductors, and the bandgap can be adjusted by strain. It makes up for the lack of band gap in graphene and brings a bright prospect for nano-electromechanical systems with mechanically adjustable optoelectronic properties.

Other language problems have been checked and modified directly in the manuscript.

Round 2

Reviewer 2 Report

The authors have properly addressed my concerns.